# Sleep-Wake and Circadian Disorders after Tick-Borne Encephalitis

**DOI:** 10.3390/microorganisms10020304

**Published:** 2022-01-27

**Authors:** Gabriele Chiffi, Denis Grandgirard, Parham Sendi, Anelia Dietmann, Claudio L. A. Bassetti, Stephen L. Leib

**Affiliations:** 1Institute for Infectious Diseases, University of Bern, 3001 Bern, Switzerland; gabriele.chiffi@ifik.unibe.ch (G.C.); denis.grandgirard@ifik.unibe.ch (D.G.); parham.sendi@ifik.unibe.ch (P.S.); 2Graduate School for Cellular and Biomedical Sciences, University of Bern, 3012 Bern, Switzerland; 3Department of Neurology, Bern University Hospital, University of Bern, 3010 Bern, Switzerland; Anelia.Dietmann@insel.ch (A.D.); Claudio.Bassetti@insel.ch (C.L.A.B.); 4Department of Neurology, Sechenov First Moscow State University, 119991 Moscow, Russia

**Keywords:** tick-borne encephalitis, sleep-wake circadian disorders, children, adults, long-term sequela, fatigue, excessive daytime sleepiness

## Abstract

Tick-borne encephalitis (TBE) is an infectious disease affecting the central nervous system. Recently, the occurrence of TBEV infections has steadily increased, reaching all-time high incidence rates in European countries. Up to 50% of patients with TBE present neurological sequelae, among them sleep–wake and circadian disorders (SWCD), which are poorly characterized. The aim of this review is to investigate the prevalence, clinical characteristics, and prognosis of SWCD after TBE. The literature review was performed in accordance with PRISMA guidelines. The quality of the paper was assessed using a standardized quality assessment. The analysis of SWCD was categorized into four different time intervals and two age groups. The literature search identified 15 studies, five including children and 10 including adults. In children, fatigue was most frequently observed with a prevalence of 73.9%, followed by somnolence/sleepiness, restlessness, and sleep-wake inversion. In adults, tiredness/fatigue was the most reported sequela with a prevalence of 27.4%, followed by extensive daytime sleepiness/somnolence, and insomnia (3.3%). Two studies showed impaired social outcomes in patients after TBE infections. SWCD after TBE in children and adults is a newly recognized sequela. Additional clinical and experimental research is needed to gain more precise insight into the clinical burden of SWCD after TBE and the underlying mechanisms.

## 1. Introduction

Tick-borne encephalitis virus (TBEV) is a positive-sense single-stranded RNA virus belonging to the Flaviviridae family [1]. Three different TBEV subtypes, differing in their virulence, have been identified, namely, the European (TBEV-Eur), Siberian (TBEV-Sib), and Far Eastern (TBEV-FE). Additionally, other tick-borne flaviviruses are found around the world, such as the Powassan virus, which has been a cause of encephalitis in the United States [2]. The number of human TBE cases has increased approximately fourfold over the last 30 years [3]. It has been hypothesized that climate change has played a role in this increase over the last decades. As such, three different aspects have been hypothesized to describe how climate change can increase the prevalence of tick-borne diseases (TBD). Firstly, the abundance of the ticks in existing populations can be increased. Secondly, changes in temperature can lead to populations of ticks spreading to higher latitudes. Lastly, with warmer weather around the year, higher tick activity can occur, which can then lead to a longer seasonal activity [4]. For tick-borne encephalitis, there are different hypotheses regarding how climate might influence the prevalence of tick-borne encephalitis in the future. There have been studies finding that increased temperatures, especially if they result in mild winters and early arrivals of spring, are positively correlated with numbers of recorded TBE cases [5,6]. On the other hand, there have been studies that did not see any significant influence of climate on the prevalence of TBE [7,8]. Some researchers have hypothesized that if temperature increases, tick-borne diseases might decrease due to changes in faunal diversity [7,9,10].

The transmission of TBEV is typically through the bite of an infected *Ixodes* tick [11]. Less frequent modes of transmission, such as through ingestion of unpasteurized milk, have also been reported [12,13]. More information beyond the scope of this review, including the epidemiology, pathogenesis, clinical manifestation, diagnostics, treatment, and prevention of tick-borne encephalitis (TBE), has been reviewed elsewhere [14,15]. The resulting research intensification allowed for the identification of various neurological and neuropsychological sequelae. These include headache, cognitive and neuropsychiatric disturbances (e.g., apathy, irritability, memory, and concentration disorders), hearing loss, vision disturbances, balance and coordination disorders, flaccid paresis, and paralysis [13,16,17,18,19,20,21,22,23]. Sleep-wake and circadian disorders (SWCD) have often been anecdotally reported, but the extent thereof and how long they persist is unknown. Therefore, in this article, we review the current knowledge regarding TBE-associated sleep-wake and circadian disorders (SWCD).

## 2. Materials and Methods

The systematic review was performed according to the PRISMA guidelines [24], and is depicted in Figure 1. Literature research and selection were performed by G.C. and A.D., including all references found up to 16 April 2021. MEDLINE and EMBASE/Ovid databases, as well as Google Scholar, were searched, using the following terms: “tick-borne encephalitis” AND “sleep disorders” OR “sleep disturbances”. Only papers in English or German with original data were considered.

Two hundred seventy-eight references were identified through the search. Duplicates were removed, and titles and abstracts were screened to ensure that the studies were within the scope of the review. In a second step, the full texts of all papers were screened. Single case reports and studies that did not report the exact number of patients with sleep-wake and circadian disorders were excluded, as they did not report a prevalence of SWCD. As the last step, the titles of the references from these selected studies were screened to assess whether any additional literature could be identified, which was, however, not the case. 

Each selected study was reviewed for (i) study population (children/adults), (ii) number of subjects, (iii) how the sequela was assessed, (iv) the observation period when the sleep-wake and circadian disorders were assessed (less than one month after TBE (acute phase), between one and six months, between six and 12 months and later than 12 months (chronic phase) and (v) for how long these disorders persisted. For the assessment of SWCD after TBE, predefined clinical criteria, as outlined in Table 1, were used.

Only a few papers provided a definition for the SWCD that was reported in the paper or how it was assessed. Therefore, each criterion that was found in the literature was first considered individually. Then, as a next step, criteria that are considered synonyms by most experts were clustered to better summarize the results of different studies. The quality of each publication was additionally assessed using a standard quality assessment criterion for evaluating primary research papers [40].

## 3. Results

Fifteen studies were included in this review and are summarized in Table 2. Overall, the quality of the assessed papers was relatively low, with a mean score of 13 (range 7–18) out of 20 possible points. The detailed quality assessment is provided in Table A1. Five studies reported sleep-wake and circadian disorders in children (mean ages ranging from 8.4 to 11.4 years), and 10 in adults (mean ages between 44.3 and 62 years). In the following paragraphs, these two categories are further explored separately. An overview of all reported sequelae clustered for synonyms and categorized by observation period is outlined in Table 3. Terms were considered synonyms in cases of an agreement among sleep experts. As such somnolence, sleepiness and extensive daytime sleepiness were all considered synonyms, as were fatigue and tiredness, as well as sleep disorders and sleep disturbances. The criteria used in the papers without clustering of synonyms are displayed in Table A2.

### 3.1. Sleep-Wake and Circadian Disorders (SWCD) in Children

Five studies including a total of 623 children described SWCD after TBE. In only one of the five studies [41] was the occurrence of SWCD followed longitudinally for 12 months. Two studies reported prevalence only in the acute phase [42,43], and two only in the chronic phase [44,45]. No studies investigating sequelae between six and 12 months were found. The numbers of included subjects ranged from eight to 371 individuals per study. The prevalence of patients reporting any form of SWCD in the three studies that investigated the acute phase was 69.1% (*n* = 387). Between 1–6 months, only one study reported SWCD in the form of fatigue in 52% of the subjects. In the chronic phase, the reported prevalence of SWCD ranged from 37.5% (*n* = 3) to 54.5% (*n* = 30) [44,45].

In the acute phase, the most frequent form of SWCD was fatigue, reported in 88.4% (*n* = 345) of the subjects [43,44]. In the chronic phase, the overall proportion of subjects with fatigue was 34.9% (*n* = 22) [44,45]. Somnolence/Sleepiness, which describes the likelihood of an individual to fall asleep, was reported in three studies in the acute phase, with a reported prevalence of 7.3% (*n* = 41) [41,42,43]. Restlessness and sleep-wake inversion were both only reported during one time period) [41,44].

In the study by Schmolck et al. [42] on 19 children with TBE, eight children reported fatigue, and somnolence was diagnosed in four children. MRI was performed in four children and revealed abnormalities in three of them. Abnormalities in the thalamus were reported in two cases [42]. The study by Engman et al. [45] also included two additional groups for comparison. Eight children with TBE were compared with 12 children with neuroborreliosis and 15 children with other pediatric diseases. At the one-year follow-up, three of the eight children with TBE reported fatigue, which was significantly higher than the proportion in the neuroborreliosis group (*n* = 0) or the control group (*n* = 1, *p* = 0.0266) [45].

**Table 2 microorganisms-10-00304-t002:** Studies included in the review reporting the presence of sleep-wake and circadian disorders after TBEV in children and adults.

Study	Study Population	Number of Subjects	Assessed Sequelae	Prevalence Acute Phase (<1 month)	Prevalence (1–6 months)	Prevalence (6–12 months)	Prevalence Chronic (>12 months)
Krbková, 2015 [41]	Children	170	Sleepiness	5.9% (*n* = 10)	n.a.	n.a.	n.a.
			Sleep-wake inversion	0.59% (*n* = 1)	n.a.	n.a.	n.a.
			Fatigue ^1^	n.a.	52% (*n* = 88)	0%	n.a.
Schmolck, 2005 [42]	Children	19	Fatigue	42.1% (*n* = 8)	n.a.	n.a.	n.a.
			Somnolence	21.05% (*n* = 4)	n.a.	n.a.	n.a.
Lešničar, 2003 [43]	Children	371	Fatigue	90.8% (*n* = 337)	n.a.	n.a.	n.a.
			Somnolence	7.2% (*n* = 27)	n.a.	n.a.	n.a.
Fowler, 2013 [44]	Children	55	Fatigue	n.a.	n.a.	n.a.	34.5% (*n* = 19)
			Restlessness	n.a.	n.a.	n.a.	20% (*n* = 11)
Engman, 2012 [45]	Children	8	Fatigue	n.a.	n.a.	n.a.	37.5% (*n* = 3)
Rezza, 2015 [46]	Adults	60 ^2^	Tiredness	38.3% (*n* = 23)	n.a.	n.a.	n.a.
			Insomnia	3.3% (*n* = 2)	n.a.	n.a.	n.a.
Czupryna, 2010 [23]	Adults	687	Sleep disorders	12.4% (*n* = 77)	n.a.	n.a.	13.2% (*n* = 5)
Mickiene, 2002 [19]	Adults	94	Fatigue	62.8% (*n* = 59)	n.a.	n.a.	Not specified ^3^
Czupryna, 2018 [47]	Adults	221 ^4^	Sleep disorders	10% (*n* = 22)	4.5% (*n* = 10)	n.a	n.a.
			Fatigue	0% (*n* = 0)	21.3% (*n* = 47)	n.a.	n.a.
Karelis, 2011 [48]	Adults	228	Somnolence	15.4% (*n* = 35)	n.a.	n.a.	n.a.
			Fatigue	44% (*n* = 44)	n.a.	n.a.	61% (*n* = 61)
			Sleep disturbances	n.a.	n.a.	n.a.	42% (*n* = 42)
Lotric- Furlan, 2017 [49]	Vaccinated adults	39	Sleep disturbances	41% (*n* = 16)	5.1% (*n* = 2)	n.a.	n.a.
	Adults	78		25.6% (*n* = 20)	7.7% (*n* = 6)	n.a.	n.a.
Hansson, 2020 [50]	Vaccinated adults	53	Fatigue	98% (*n* = 52)	79% (*n* = 33) (1–3 mo.) 87% (*n* = 13) (3–6 mo.)	44% (*n* = 8)	n.a.
Bogovic, 2018 [51]	Adults	420	Fatigue	n.a.	21.4% (*n* = 65)	15.9% (*n* = 33)	6.4% (*n* = 27)
			Sleep disorders	n.a.	2.6% (*n* = 8)	2.9% (*n* = 6)	2.9% (*n* = 12)
Veje, 2016 [52]	Adults	96	Fatigue	74% (*n* = 71)	65% (*n* = 52) (0.5–2 mo.) 58% (*n* = 21) (3–6 mo.)	n.a.	n.a.
Veje, 2021 [53]	Adults	22	Fatigue	n.a	n.a.	n.a	31.8% (*n* = 7)
			Excessive Daytime Sleepiness	n.a.	n.a.	n.a.	54.5% (*n* = 12)

^1^ Exact percentage of fatigue was not provided but provided in a category with headache. ^2^ 367 cases were analyzed but only for 60 cases were sequelae reported. ^3^ Exact percentage of fatigue was not provided, but only listed as a common complaint. ^4^ 1072 cases were assessed, but only 221 developed sequelae.

**Table 3 microorganisms-10-00304-t003:** Prevalence for each sequela per timepoint with synonyms aggregated.

	Assessed Sequela	Prevalence Acute (<1 month)	Prevalence between 1 and 6 Months	Prevalence between 6 and 12 Months	Prevalence Chronic (>12 months)	Total per Sequela
Children	Fatigue [41,42,43,44,45]	88.5% (*n* = 345)	51.8% (*n* = 88)		37.9% (*n* = 22)	73.9% (*n* = 455)
	Restlessness [44]				20% (*n* = 11)	20% (*n* = 11)
	Sleep-wake inversion [41]	0.6% (*n* = 1)				0.6% (*n* = 1)
	Somnolence/Sleepiness [42,43]	7.3% (*n* = 41)				7.3% (*n* = 41)
	Total per time	34.5% (*n* = 387)	51.8% (*n* = 88)		29.2% (*n* = 33)	
Adults	Excessive Daytime Sleepiness [48,53]	15.3% (*n* = 35)			54.5% (*n* = 12)	18.8% (*n* = 47)
	Fatigue [19,46,47,48,50,51,52,53]	39.9% (*n* = 249)	26.9% (*n* = 231)	18.1% (*n* = 41)	17.5% (*n* = 95)	27.4% (*n* = 616)
	Insomnia [46]	3.3% (*n* = 2)				3.3% (*n* = 2)
	Sleep disorders/Sleep disturbance [23,47,48,49,51]	13.2% (*n* = 135)	4.0% (*n* = 26)	2.9% (*n* = 6)	10.7% (*n* = 59)	9.3% (*n* = 226)
	Total per time	21.7% (*n* = 421)	17.1% (*n* = 257)	10.8% (*n* = 47)	14.8% (*n* = 166)	

### 3.2. Sleep-Wake and Circadian Disorders (SWCD) in Adults

Ten studies including 2037 adults assessed SWCD after TBE. Nine studies had a retrospective and one study a prospective design [51]. The majority of the studies assessed self-reported symptoms. Only one study used objective measurement instruments (polysomnography) [53]. The quality of the studies was rated as poor in seven of the 10 included publications. Overall, the participants from the studies ranged from 22 to 687 individuals per study.

None of the studies investigated SWCD during all four observation periods defined in this review. Two studies investigated sequelae for three consecutive observation periods. Both of them reported SWCD between one and six months, and between six and 12 months. One of the two studies additionally reported SWCD in the acute phase [50], while the other reported findings in the chronic phase [51]. Six studies investigated SWCD at two observation periods. Three studies reported SWCD during the acute phase, as well as between one and six months [47,49,52], while three other studies reported results in the acute and chronic phase [19,23,48]. Two studies reported SWCD only for one observation period. One study reported SWCD in the acute phase [46], while the other did in the chronic phase only [53]. 

The prevalence of patients with any form of SWCD was 21.7% (*n* = 421) in the acute phase. Between 1–6 months 17.1% (*n* = 257) of study participants reported SWCD. In the period between six and 12 months, SWCD-related sequelae were reported in 10.8% (*n* = 47) of patients. The latter proportion derives mainly from two studies with relatively high differences in the number of patients (18 and 208, respectively) [50,51]. In the chronic phase, the reported prevalence across all studies was 14.8% (*n* = 166). 

Overall, four types of SWCD were observed, including fatigue/tiredness, sleep disorders/disturbances, insomnia and somnolence/extensive daytime sleepiness (EDS). These SWCD were reported in every observation period across the studies. Fatigue/tiredness and sleep disorders/disturbances were consistently reported. Sleep disturbances, which generically describes difficulties in falling asleep and maintaining sleep, were reported for three periods. 

Fatigue/tiredness was reported over all observation periods in 27.4% (*n* = 616) of subjects. During the acute stage, it was observed in 39.9% (*n* = 249) of the assessed subjects in studies that specifically investigated this sequela. Between 1–6 months fatigue/tiredness was described in 26.9% (*n* = 231) of subjects, while in the period between six and 12 months 18.1% (*n* = 41) of subjects reported fatigue or tiredness. In the chronic phase, 17.5% of the subjects reported this complaint (*n* = 95). 

When considering all reports independently of the timepoint, sleep disorders/disturbances, which refers to generically different disorders affecting night-time sleep with difficulties falling asleep or maintaining sleep as well as daytime activity with difficulties staying awake, or a combination of both, were reported in 9.3% (*n* = 226) of all subjects. Sleep disorders/sleep disturbances occurred in 13.2% (*n* = 135) of subjects during the acute phase. Between 1–6 months and 6–12 months sleep disorders/sleep disturbances were described in 4.0% (*n* = 26) and 2.9% (*n* = 6) of subjects respectively. In the chronic stage, sleep disorders/sleep disturbances were observed in 10.7% (*n* = 59) of subjects.

Two studies with a good quality score are explored in further detail. A prospective observational study on long-term outcomes after TBE in Slovenia published in 2017 included 420 TBE patients from 2007 to 2012 [51]. After six months 304 patients, after 12 months 207 patients, and after 2–7 years 420 patients were seen for follow-up visits. For long-term follow-up visits, 295 control subjects were also included [51]. The control group was age- but not sex-matched. Long-term SWCD were investigated by means of a clinical interview, neurological examination, and questionnaires. A post-encephalitic syndrome was described in this study in 42% of patients six months after acute illness, and still persisting in 33% of patients 2–7 years after acute TBE. While 21.4% of the TBE patients reported fatigue after six months, this proportion decreased to 15.9% at 12 months, and 6.4% at long-term follow-up. Sleep disorders were reported by 2.6% of the patients at six months, and 2.9% after 12 months or later. Frequent residual symptoms in the long-term follow-up were headache (31%), memory and/or concentration disorders (29%), emotional lability (14%), arthralgias and/or myalgias (14%), and dizziness (8%). This study found no significant difference regarding fatigue and SWCD between the patient population and the control group at the later follow-up [51].

In a study by Veje et al., 92 patients were assessed and compared to 58 age- and sex-matched controls from the same residential area in Sweden. This study used the Functional Outcome of Sleep Questionnaire (FOSQ) and assessed the time-course of fatigue over six months after admission for TBE. While 74% of patients complained of fatigue at admission at the hospital, this percentage decreased to 65% between 0.5 and two months, and to 58% when assessed between three and six months. At a long-term follow-up ranging from two to 15 years (mean 5.5 years) after admission, a significant difference in fatigue was reported between TBE patients compared to the control group, assessed by the mean value of the FOSQ in tiredness and fatigue. Furthermore, a significant difference was described in the social outcome of the FOSQ between the TBE patients and the controls [52]. 

A more recent study by Veje et al included polysomnography. Twenty-two patients with tick-borne encephalitis were included in the study to assess SWCD one to five years after the initial hospital admittance. The authors found that the patient group reported significantly more complaints of fatigue (31.8%) compared to the control group (*n* = 20, 5.3%, *p* < 0.05). This effect was also seen in two standardized questionnaires (Epworth Sleepiness Scale and the Functional Outcome of Sleep Questionnaire). However, no difference in the macrostructure of the sleep could be identified by polysomnography between the control group and the TBE patient group [53].

## 4. Discussion

This literature review highlights that SWCD after TBE are frequent and represent a considerable clinical burden. 

Only 15 studies covering this entity were found in our literature review, five of which (two with children and three with adults) could be rated with moderate to high quality. The results of this literature review are in agreement with a previous review by Haglund and Günther [21] suggesting that SWCD consecutive to TBE is under-investigated and likely underreported.

The clinical burden of SWCD after TBE impairs the quality of life. Veje et al. reported that patients with a history of TBE scored significantly lower than controls in questionnaires investigating the dimensions of memory and learning, executive function, vigilance, and physical impairment after a median follow-up of 5.5 years. These findings, together with reduced motivation and the higher likelihood to fall asleep throughout the day, were also associated with lower scores in questionnaires evaluating dimensions of social outcomes [52,53]. Lower scores in the questionnaires were also associated with controls having obstructive sleep apnea [53].

The true prevalence of SWCD after TBE is difficult to assess from data in the literature, since very few studies have used a longitudinal prospective design or a standardized measurement tool at different observation periods after TBE. The majority of studies assessed symptoms in the acute phase. Therefore, differentiation between recovery from the acute disease and the onset of long-lasting or permanent sequelae is clinically difficult. Nonetheless, SWCD in children persisted in up to one-third of subjects for 12 or more months [44,45]. Moreover, fatigue was significantly more pronounced after TBE compared to children with neuroborreliosis or other pediatric diseases [45]. The same trend was seen for SWCD after TBE in adults. The proportion of SWCD after TBE decreased over time. However, the wide range of reported prevalence at 12 or more months (2.9% to 61%) does not allow for any firm conclusion on the true long-term prevalence of SWCD after TBE.

This literature review also demonstrated that SWCD after TBE are frequently poorly defined. Terms such as “fatigue” or “sleep disturbances” were mostly used without a precise definition. The reported investigations often lacked objective parameters. Fatigue and sleep-wake disturbances were assessed by the use of standardized questionnaires in only 13% of the included publications.

With respect to the pathophysiology underlying SWCD, brain correlates for these problems have been suggested. An MRI investigation of 102 TBE patients revealed lesions in 18 of them, 15 of which had abnormalities confined only to the thalamus. Three of the 18 patients had abnormalities in the thalamus with additional lesions in the cerebellum, the brainstem, and the caudate nucleus [54]. Another study assessed MRI of 25 patients who experienced vaccination breakthrough and suffered from a TBE infection after a TBE vaccination. The authors identified nine patients in this cohort with pathological findings. In seven patients the affected area was the diencephalon, with involvement of the thalamus and the basal ganglia. Additionally, the brainstem with changes in the pons, the medulla oblongata, and the midbrain were affected in four patients [50]. Since the thalamus plays an important role in sleep oscillation and therefore influences sleep onset, stabilization, and termination [55], the damage observed in this structure after TBE implies possible SWCD, including the occurrence of fatigue in the affected patients. This research, together with findings from studies that showed a significantly higher percentage of patients reporting SWCD compared to a control group [45,51,52,53], indicates that SWCD are specific sequelae after TBE. For future research, it would therefore be important to assess potential damages to the thalamus and correlate them to the observed symptoms.

In sum, uncertainties remain concerning the frequency and impact of SWCD after TBE. In view of the effect on the patient’s quality of life, SWCD should be studied more systematically. The use of precise term definitions, the implementation of objective measures such as polysomnography, sleep latency tests, and actigraphy, and the inclusion of adequate control groups, combined with validated questionnaires, are needed to obtain a more accurate picture of TBE-related SWCD. Moreover, studies should be performed in a prospective and longitudinal manner with sequential examinations. Finally, to gain more insight into how TBEV infection affects sleep-wake mechanisms, molecular studies and non-invasive imaging methods (e.g., MRI), as well as experimental models, should be further developed and investigated.

## 5. Conclusions

SWCD after TBE persist in children (between 37.5 and 54.5%) and adults (14.8%) for more than 12 months and significantly impair the quality of life of the patients. However, post-TBE SWCD are still under-investigated and underreported. Additional epidemiological, clinical and experimental research conducted by interdisciplinary teams is needed to understand the exact frequency, impact and underlying pathophysiology of this group of disturbances.

## Figures and Tables

**Figure 1 microorganisms-10-00304-f001:**
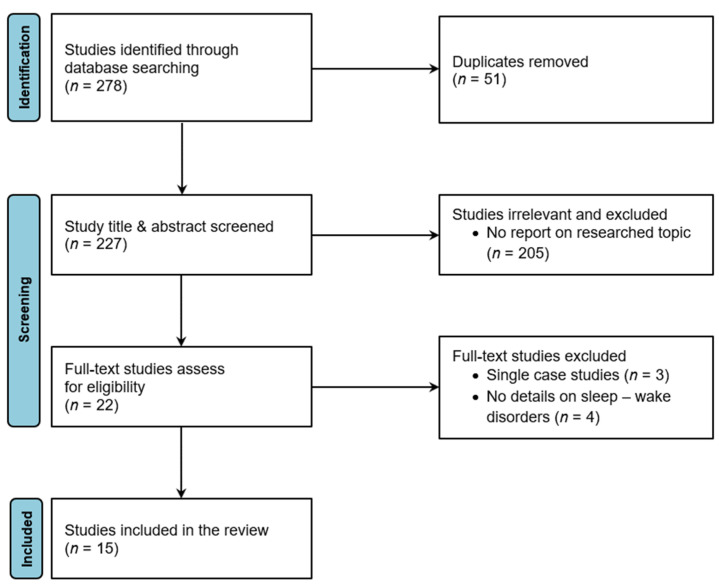
Flowchart describing search and selection strategy.

**Table 1 microorganisms-10-00304-t001:** Definitions of sleep-wake and circadian disorders.

Phenomena	Description	Ref.
Sleep–wake disorders (SWD)/sleep disorders or disturbances	Generic term describing different disorders that may affect either night-time sleep, with difficulties falling asleep or maintaining sleep, or daytime, with difficulties staying awake, or a combination of both. These impairments manifest as frequent waking at night, excessive fatigue during the day, problems with sleep-wake schedule, nocturnal wandering and in general impaired function of sleep or sleep stages or partial arousals during sleep, potentially resulting in non-refreshing sleep. Causes of SWD are manifold, from sleep-related breathing disorders to neurological and psychiatric disorders.	[25,26,27,28,29]
Fatigue	Complaint of physical and/or mental exhaustion with difficulties in initiating or sustaining voluntary activities that are not significantly improved by increased rest or sleep. Subjective measurement by questionnaires such as the Fatigue Severity Scale (FSS), no objective measurement available.	[30,31]
Excessive Daytime Sleepiness (EDS)	Subjective difficulty maintaining wakefulness or alertness during major waking episodes of the day, with involuntary/unintentional and involuntary naps in monotonous situations, acquired need of scheduled napping during the day. Subjective measurement by questionnaires such as the Epworth Sleepiness Scale (ESS), objective measurement by multiple-sleep-latency test (MSLT).	[26,32,33]
Hypersomnia	An objective complaint of excessive sleep need. An objective assessment of an excessive quantity of sleep: at least 10 h of sleep duration over 24 h of the day with the nocturnal component providing at least 9 h of sleep duration. Objective measurement by actigraphy and/or polysomnography (PSG) ad libitum. More recent definitions of idiopathic hypersomnia adjusted the threshold to a sleep time >660 min in a 24-h cycle or a mean sleep latency in MSLT of ≤8 min.	[26,33,34]
Circadian rhythm disorders (CD)	Chronic or recurrent sleep disturbances due to alteration of the circadian system and/or misalignment between the environment and an individual’s sleep–wake cycle. Leads to advanced or delayed sleep phases up to day/night sleep inversion, interfering with social and work life requirements.	[35,36]
Insomnia	Difficulties initiating or maintaining sleep, even under appropriate circumstances and opportunity to sleep and without known cause of sleep disturbance (e.g., sleep-related breathing disorder, restless leg syndrome etc.). Furthermore, daytime consequences with impairment of social life or work are requirements for the diagnosis of insomnia. Subjective measurement by questionnaires such as the Insomnia Severity Scale (ISS).	[26,34]
Sleepiness/Somnolence	Describes a phenomenon that can be a symptom of medical, psychiatric, neurological or primary sleep disorders as well as a normal physiological state which is observed over a 24-h period in humans. It is most often defined as the tendency of an individual to fall asleep, which is also referred to as sleep propensity. Alternatively, it has been stated that sleepiness can be defined as an abnormal behavior when it either occurs at inappropriate time or when it is not desired.	[30]
Restlessness	Restlessness has been historically difficult to describe and has lacked a clear definition. Some literature defines restlessness as a state of aimless and poorly organized motor activity that stems from physical or mental unease. Other research papers use the term restlessness to measure sleep quality by asking subjects whether they suffer from disturbed sleep/insomnia. With respect to restlessness during sleep, this has previously been defined to be characterized by persistent or recurrent movements of the body, the presence of arousals and being briefly awakened during sleep.	[37,38,39]

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
