# Peer review of "Sleep-Wake and Circadian Disorders after Tick-Borne Encephalitis"

_microorganisms, 2022, doi:10.3390/microorganisms10020304_

Round 1

Reviewer 1 Report

The manuscript is interesting in that it draws attention to an understudied clinical outcome that warrants further investigation. A major weakness, through no fault of the authors, is the limited number of studies included in this review, as well as major differences in study methods and variables of interest in selected studies. This is likely why a deeper meta-analysis wasn’t performed. Regardless, the information is presented in a coherent fashion and is appropriate to include as a short review.  

- there is a recurring “Error! Reference source not found.’ Message occurring throughout. Please correct before moving forward in review process. See - Ln  52, 69,  79, 86, 91.

Author Response

Comment 1: The manuscript is interesting in that it draws attention to an understudied clinical outcome that warrants further investigation. A major weakness, through no fault of the authors, is the limited number of studies included in this review, as well as major differences in study methods and variables of interest in selected studies. This is likely why a deeper meta-analysis wasn’t performed. Regardless, the information is presented in a coherent fashion and is appropriate to include as a short review

Answer: We appreciate this feedback of the reviewer. We agree that a meta-analysis would have rounded off the review nicely, but as assumed by the reviewer due to the limited number of studies and their methodological differences this was not feasible. We however very much appreciate the reviewer’s acknowledgement of our efforts to put forth a coherent review of the existing literature.

Comment 2: - there is a recurring “Error! Reference source not found.’ Message occurring throughout. Please correct before moving forward in review process. See - Ln  52, 69,  79, 86, 91.

Answer: We thank the review for mentioning these formatting errors. As this point was raised by both reviewers, we checked our submission and found that this issue was not present in the version which we uploaded for review. Several changes to the layout of the paper were made by the journal after our submission, and we therefore assume that during this step changes leading to this were made.

Measures. We have reformatted the Figure legends and cross references. In addition we have uploaded the revisions as a pdf to ensure that there are no unwanted changes in the process of conversion into a pdf file.

Reviewer 2 Report

This is a systematic review of literature about circadian disorders following TBE in children and adults. The study is interesting and topic is emerging and pertinent. I think the paper would contribute to the existing literature. Some suggestions to authors to improve the paper before publications are the following: 

1. Introduction line 34 - It should be added that TBEV it is related to Powassan virus, present in North America.

2. Introduction line 37-42 Authors should add few sentences about influence of climate change on the TBEV vector, Ixodes tick and how the incidence of the disease is expected to increase in the coming years. For example, one recent study from USA examined the influence of climate change on the incidence of Lyme disease ( TBEV, POWV and LD share the same vector Ixodes tick) and found that the incidence will increase for 20% in coming decades due to climate change ( https://pubmed.ncbi.nlm.nih.gov/30473737/)

3. Materials and Methods line 52- please revise. Additionally there are several other spaces in the text where similar text appears . I am shocked that authors allowed this to happen. 

4. Line 223- "after the vaccination"? Which vaccine, and how is that related to TBE?

5. In discussion authors should dedicate a paragraph about potential pathophysiology behind these disturbances and provide the framework and hypothesis  for future research. 

6. Methodology- why 3 case reports were excluded? Just for being case reports or they did not provide enough information etc?

7. One of the essential steps in PRISMA methodology is to screen the references of articles selected through database search in order to pick up papers that might have been missed by initial database search. Authors should report if they did it, and if any additional reference was found?

Author Response

Comment 1: Introduction line 34 - It should be added that TBEV it is related to Powassan virus, present in North America.

Answer: We thank the reviewer for suggesting options to add more geographical locations with related disorders.

Measure: As such in Line 36 the following sentence was added: Additionally, other tick-borne flaviviruses are found around the world, such as the Powassan virus, which has been a cause of encephalitis in the United States [2].

Comment 2: Introduction line 37-42 Authors should add few sentences about influence of climate change on the TBEV vector, Ixodes tick and how the incidence of the disease is expected to increase in the coming years. For example, one recent study from USA examined the influence of climate change on the incidence of Lyme disease ( TBEV, POWV and LD share the same vector Ixodes tick) and found that the incidence will increase for 20% in coming decades due to climate change ( https://pubmed.ncbi.nlm.nih.gov/30473737/)

Answer: We appreciate the reviewer’s suggestion of introducing the potential effect of climate on the tick-borne encephalitis and providing appropriate literature.

Measure: Using this and additional literature a section has been added addressing this topic. This section starts on Line 39 and entails the following:

It has been hypothesized, that climate change has played a role in this increase over the last decades. With warmer winters a higher rate of ticks survives and there has been evidence that with more humid and warmer summers the tick activity is increased through-out the years. There are different hypotheses, how climate might influence tick activity and the development of tick-borne diseases in the future. Some researchers even hypothesize, that if temperature increases, the tick-borne diseases might decrease, due to changes in faunal diversity [4]–[6].

Comment 3: Materials and Methods line 52- please revise. Additionally there are several other spaces in the text where similar text appears . I am shocked that authors allowed this to happen. 

Answer: We thank the review for mentioning these formatting errors. As this point was raised by both reviewers, we checked our submission and found that this issue was not present in the version which we uploaded for review. Several changes to the layout of the paper were made by the journal after our submission, and we therefore assume that during this step changes leading to this were made.

Measure: We have reformatted the Figure legends and cross references and have uploaded the revisions as a pdf to ensure that there are no unwanted changes.

Comment 4: Line 223- "after the vaccination"? Which vaccine, and how is that related to TBE?

Answer: We appreciate the reviewers request for more details, as we assumed that it was clear which vaccination we refer to. The vaccination which we refer to is the TBE vaccination.

Measure: This information has now explicitly been added. Line 250 now reads: Another study assessed the MRI of 25 patients who had vaccination breakthrough and suffered from a TBE infection after a TBE vaccination.

Comment 5: In discussion authors should dedicate a paragraph about potential pathophysiology behind these disturbances and provide the framework and hypothesis  for future research. 

Answer: We appreciate the authors suggestion to include this paragraph. As such we restructured part of the discussion to structure this section by using previously included text of the discussion. Additional importance was added to this section by adding a statement about the importance for future research. Lastly the section was added almost to the end of the discussion to highlight the importance thereof for future research.

Measure: The section with starts with line 246 now reads: With respect to the pathophysiology underlying the SWCD brain correlates for these problems have been suggested. An MRI investigation of 102 TBE patients revealed lesions in 18 of them, 15 of which had abnormalities confined only to the thalamus. Three of the 18 patients had abnormalities in the thalamus with additional lesions in the cerebellum, the brainstem, and the caudate nucleus [52]. Another study assessed the MRI of 25 patients who had vaccination breakthrough and suffered from a TBE infection after a TBE vaccination. The authors identified nine patients in this cohort with pathological findings. In seven patients the affected area was the diencephalon, with an involvement of the thalamus and the basal ganglia. Additionally, the brainstem, with changes in the pons, medulla oblongata, and the midbrain were affected in four patients [48]. Since the thalamus plays an important role in sleep oscillations and therefore influences sleep onset, stabilization, and termination [53], the damage observed in this structure after TBE im-plies possible SWCD including the occurrence of fatigue in the affected patients. This re-search, together with the findings from the studies which showed a significantly higher percentage of patients reporting SWCD compared to a control group [43], [49]–[51], indicates that SWCD is a specific sequela after TBE. For future research, it would therefore be important to assess potential damages to the thalamus and correlate them to the observed symptoms.

Comment 6: Methodology- why 3 case reports were excluded? Just for being case reports or they did not provide enough information etc?

Answer: One of the major aims of this study was to summarize studies which report on the prevalence of the SWCD. As these case reports do not report prevalence as the mainly were an overview of single cases, they were not considered. Additionally, in our recollection, most case studies tended to be written in a rather narrative style which did not provide sufficient scientific information.

Measure: A statement explaining this step has now been added on Line 70: as they did not report a prevalence of SWCD.

Comment 7: One of the essential steps in PRISMA methodology is to screen the references of articles selected through database search in order to pick up papers that might have been missed by initial database search. Authors should report if they did it, and if any additional reference was found?

Answer: Yes, this step was conducted as the titles of the references were screened, however no new literature was found. We assume that this is due to the very broad search terms which were used.

Measure: A statement on this procedure has now been added in Line 71: “As a last step the titles of the references of the selected studies were screened to assess whether any additional literature can be identified, which was however not the case.

Round 2

Reviewer 2 Report

References 4-6 do not represent the literature on the climate change influence of tick borne infection. Ref 5 is more than 20 years old. I would ask authors to elaborate more on climate change influence on tick borne diseases and cite appropriate and up to date literature

Author Response

Response: “We appreciate the reviewer’s comment and have adapted the section. We included a new section which describes the modalities how climate may modulate tick-borne diseases and included the relevant and up-to-date literature. 

We have addressed the issue with the formatting of the references, which lead to incorrect displays of references in the bibliography.